# A Method for Rotor Speed Measurement and Operating State Identification of Hydro-Generator Units Based on YOLOv5

**Jiajun Liu, Lei Xiong, Ji Sun \*, Yue Liu, Rui Zhang and Haokun Lin**

School of Electrical Engineering, Xi'an University of Technology, Xi'an 710000, China
\* Correspondence: 1211911003@stu.xaut.edu.cn; Tel.: +86-185-2174-4707

**Abstract:** With the rapid development of artificial intelligence, machine vision and other information technologies in the construction of smart power plants, the requirements of power plants for the state monitoring of hydro-generator units (HGU) are becoming higher and higher. Based on this, this paper applies YOLOv5 to the state monitoring scenario of HGU, and proposes a method for rotor speed measurement (RSM) and operating state identification (OSI) of HGUs based on the YOLOv5. The proposed method is applied to the actual RSM and OSI of HGUs. The experimental results show that the Precision and Recall of the proposed method for rotor image are 99.5% and 100%, respectively. Compared with the traditional methods, the online image monitoring based on machine vision not only realizes high-precision RSM and the real-time and accurate determination of operating states, but also realizes video image monitoring of the rotor, the operation trend prediction of the rotor and the early warning of abnormal operating states, so that staff can find the hidden dangers in time and ensure the safe operation of the HGU.

**Keywords:** artificial intelligence; hydro-generator unit; YOLOv5; rotor speed measurement; online monitoring

## 1. Introduction

An HGU is a complex coupling system of hydraulic, mechanical, and electrical systems. As the service life of the units increases, the issues of structural fatigue and deterioration become increasingly prominent. At present, large and medium-sized hydropower plants are developing an unmanned management mode and one with few personnel on duty, and the equipment maintenance method is gradually transitioning from regular preventive maintenance based on time to predictive maintenance based on state monitoring [1–3]. How to accurately monitor the unit, judge its operation state, and detect unit operation problems on time is an important issue in the state maintenance of HGU. The monitoring parameters (nonelectrical quantity) of HGUs can generally be vibration, noise signals, temperature, and rotor speed [4–6]. The rotor speed of the HGU can reflect both the state and frequency of the HGU, which is a very important detection quantity in HGU monitoring. Therefore, accurately calculating the rotor speed of the HGU is of great significance for monitoring and judging the state of HGU [7,8].

At home and abroad, the methods of RSM for HGUs include direct method and indirect method. The indirect method mainly converts mechanical rotation into other physical quantities, and converts physical quantities into velocity quantities according to the corresponding calculation formula. The mainstream indirect method is PT residual pressure velocimetry. As the name implies, the direct method measures the mechanical rotation of the object directly through the corresponding sensor. The mainstream method installs a toothed disc on the main shaft of the HGU, so that the toothed disc is connected with the main shaft of the HGU. The rotation of the main shaft of the HGU will drive the toothed disc to rotate synchronously. The toothed disc sensor will collect the corresponding pulse signal, and the current speed value of the HGU will be calculated through the processing

of the signal by the single-chip microcomputer in the later stage [9]. The above methods all have certain drawbacks. On the one hand, they cannot achieve visual monitoring of speed and cannot reproduce the accident development process afterwards. On the other hand, different speed measurement methods have varying degrees of coupling with the rotor, reducing the robustness of the measurement. The advantages and disadvantages of the commonly used classical rotor speed measurement methods are shown in Table 1.

**Table 1.** Comparison of advantages and disadvantages of rotor speed measurement methods.

| RSM Method | Advantages | Disadvantages |
|---|---|---|
| Toothed disc [10] | High measurement accuracy, Strong real-time performance, Strong anti-interference ability | The need to fix the processed toothed disc will change the spindle structure |
| PT residual pressure [11] | Able to obtain the voltage of the generator outlet PT, commonly used for electromagnetic measurement | Electrical faults and abnormal residual voltage of the primary equipment can cause inaccurate speed measurement |
| Photoelectric encoder [12] | High measurement accuracy, fast response | Susceptible to signal noise, contact type speed measurement requires coaxial installation |
| Laser Doppler [13] | Noncontact speed measurement without changing the spindle structure | High price, poor immunity |
| Machine Vision [14] | Noncontact speed measurement without changing the spindle structure, visualization of accident process | Limited usage scenarios |

In recent years, machine vision, as a hot technology, has provided effective technical support for promoting the construction of smart grids. Research based on machine vision is constantly emerging and has been validated in practical power engineering applications [15–17]. Ref. [18] adopts an intelligent detection method for transmission line defects based on reparameterized YOLOv5, which solves the problem of slow edge reasoning caused by low computing power and low memory of power patrol edge equipment. Ref. [19] applies YOLOv5, which combines the weighted bidirectional feature pyramid (BiFPN) structure to the identification of power switch cabinet state lights, assigns different weights to the feature layer to transmit more effective feature information, and solves the problem of small target recognition caused by the high-density layout of state lights. Ref. [20] proposes a method of fan blade detection and spatial positioning based on the lightweight YOLOv5. ShuffleNetv2 is used as the feature extraction backbone network to achieve accurate positioning of fan blade tip. Ref. [21] uses a method of making fused image data set to solve the problem of the small number of defective insulator samples in the insulator image data set taken by UAV aerial photography. To sum up, machine vision has been widely used in power systems, but there is relatively little research on it in hydropower plants [22,23].

The RSM method for HGUs based on image processing is a type of indirect RSM method. This method mainly tracks the target in the measured structure video captured by the camera to obtain the motion trajectory of the measurement point in the image, and then determines the motion information of the structure through the geometric relationship between the image and the real world. Unlike the contact displacement monitoring method, which requires the installation of fixed support points on the structure, the camera is installed at a fixed point far from the measured object and does not have a coupling

relationship with other electrical equipment, contributing to the non-interference of the RSM process and unit operation, greatly improving the anti-interference ability of RSM.

This paper applies the YOLOv5 algorithm to the state monitoring scenario of HGU, and proposes a method for rotor speed measurement and operation state recognition of HGU based on YOLOv5 combined with the period measurement method. A monitoring system for the rotor speed of HGU is also developed. This system not only achieves precise measurement of rotor speed, makes accurate judgments on the operating state of HGU, predicts rotor operating trends and alerts abnormal operating states, but also records real-time operating images of HGU, providing data sources for post analysis. The system developed in this paper can effectively ensure the safe and stable operation of HGUs, laying a theoretical foundation for the future development of more functional monitoring systems.

## 2. YOLOv5 Model Analysis

Object detection is a machine vision technology that can recognize semantic objects in images and provide their positions and categories. Traditional object detection methods typically include three steps: region selection, feature extraction, and feature classification. After the emergence of deep learning, object detection methods have enhanced the accuracy of feature classification and improved the efficiency of region selection, becoming a common method at present. There are two types of object detection methods: single-stage etection and two-stage detection. Single-stage detection integrates target classification, boundary localization, and feature extraction into a network, constructs end-to-end training methods, and uses regression to obtain the position of the target, reducing the repetitive calculation of image feature extraction steps. The main algorithms include SSD, YOLO, etc. Two-stage detection is based on constructing a deep convolutional neural network to extract target features, and then achieving target detection through image segmentation and positioning. The main algorithms include Faster RCNN, Fast R-CNN, SIFT, etc. [24].

The YOLO series algorithm is a target detection method based on regression thinking, which can directly predict the category and position of the target from the image without the need for candidate boxes or other intermediate steps. The advantage of this series of algorithms is its fast speed, which is suitable for real-time scenes. The YOLOv5 model has the advantages of fast reasoning speed, high precision and small model size, which makes it highly popular in the field of target detection. Figure 1 shows the overall block diagram of the YOLOv5 target detection algorithm. For a target detection algorithm, we can usually divide it into four general modules, specifically including Input, Backbone, Neck and Head, corresponding to the four red modules in Figure 1.

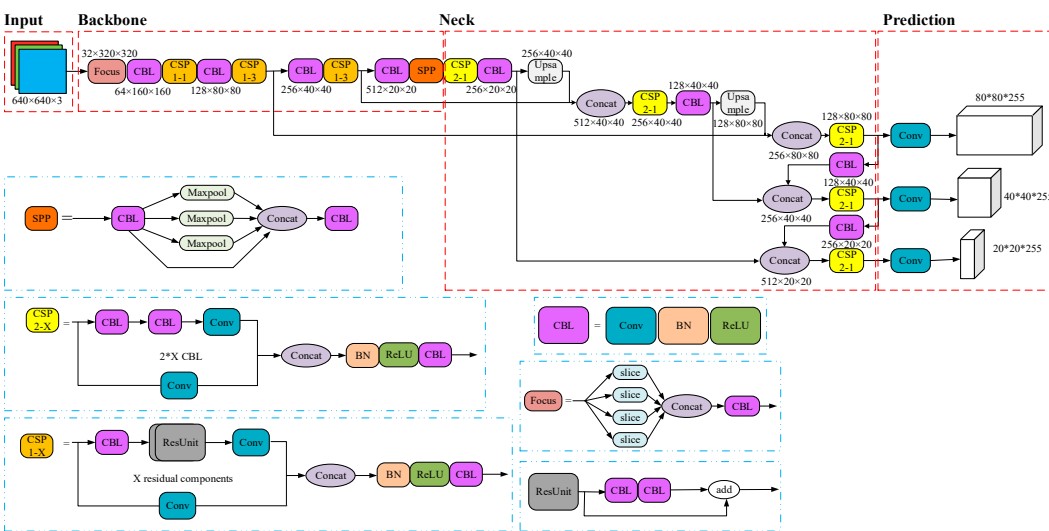

**Figure 1.** YOLOv5 model architecture diagram.

The Input represents the input image, which typically includes an image preprocessing stage, which scales the input image to the input size of the network and performs operations such as normalization. In the network training phase, YOLOv5 uses Mosaic data enhancement operations to improve the training speed of the model and the accuracy of the network. An adaptive anchor frame calculation and adaptive image zooming method are proposed. The Backbone is usually a network of classifiers with excellent performance, and this module is used to extract some common feature representations. YOLOv5 not only uses the CSPMarket53 structure, but also uses the Focus structure as the Backbone. The Neck network is usually located in the middle of the reference network and the header network, and it can be used to further enhance the diversity and robustness of features. Although YOLOv5 also uses the SPP module and FPN + PAN module, the implementation details are somewhat different. The Head is used to complete the output of target detection results. For different detection algorithms, the number of branches at the output end varies, usually including a classification branch and a regression branch. YOLOv5 leverages GIOU_Loss replaces the Smooth L1 Loss function to further improve the detection accuracy of the algorithm.

In order to evaluate the effectiveness and feasibility of the YOLOv5 model test results, in practical applications, Precision, Recall, Average Precision (AP) and mean Average Precision (mAP) are usually used as evaluation indicators [25]. The formula of the above indicators is as follows. The Precision represents the true correct proportion in the correct classification, and the Recall represents the proportion of the correct samples in the given correct samples.

$$\text{Precision} = \frac{\text{TP}}{\text{TP} + \text{FP}} \tag{1}$$

$$\text{Recall} = \frac{\text{TP}}{\text{TP} + \text{FN}} \tag{2}$$

$$\text{AP[Class]} = \sum_{i \in \text{confidence}} \text{Precision}_i[\text{Recall, Class, IOU}] \tag{3}$$

$$\text{mAP} = \frac{1}{N} \sum \text{AP}_i \tag{4}$$

In the formula, TP represents the number of tags that are positive samples and classified as positive samples. FP indicates the number of negative samples but classified as positive samples. FN indicates the number of positive samples but classified as negative samples.

Under a fixed Intersection Over Union (IOU), a given target will obtain different Precision and Recall values according to different confidence levels. Through interpolation of Precision and Recall, the continuous curve generated is the Precision–Recall (PR) curve. The AP represents the comprehensive performance of the model under different confidence levels by the area enclosed by the PR curve of a given target category and the horizontal and vertical coordinates [26]. The higher the AP value, the better the detection performance of the model. Each IOU corresponds to a different AP. The AP@.5 represents the AP when the IOU is taken as 0.5, and the AP@.5:.95 represents the average AP when the IOU is taken as 0.5 to 0.95, in steps of 0.05.

## 3. Analysis of RSM Principle

### 3.1. RSM by Period Method

The RSM principle based on image recognition technology is similar to that of digital circuit speed measurement. The digital circuit speed measurement is to count the known frequency high-frequency clock pulse with a counter within the interval of two adjacent output pulses, and then calculate the speed, which is called T-method speed measurement [27]. The schematic diagram of the period measurement method based on image recognition technology is shown in Figure 2. The time of one revolution of the HGU can

be obtained by multiplying the frame rate of the camera by the number of frames taken during one revolution of the HGU. The key is to calculate the time difference of one revolution. In the continuous frames with markers (that is, within the irradiable range of the camera), the appearance and disappearance of markers are the two time points we focus on (the judgment of key frames in the program is shown in Figure 3). The time difference corresponding to the adjacent frames that appear or disappear is the time required for the rotor to rotate for one cycle, that is, the speed of the unit. Therefore, when the unit rotates for one and a half periods, we can obtain the real-time rotor speed of the two units.

$$n = \frac{60}{f_0 \times M} = \frac{60}{f_p} = \frac{60}{f_q} \tag{5}$$

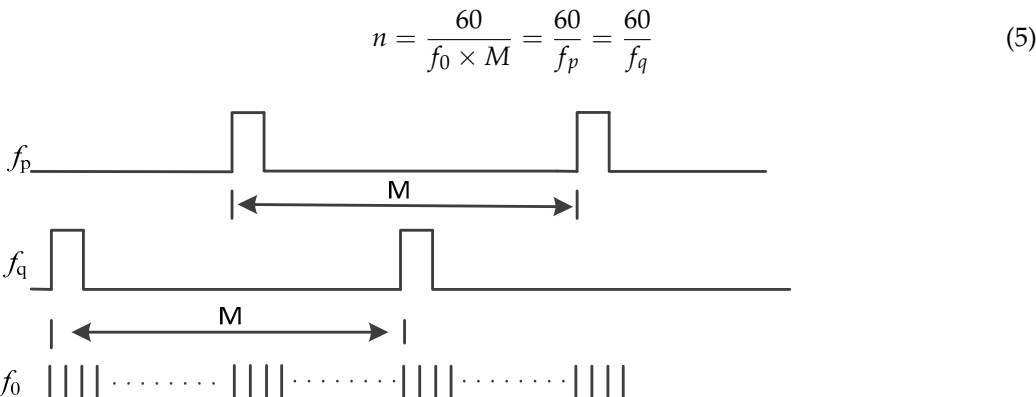

**Figure 2.** Waveform of velocity measurement principle of video measurement period method.

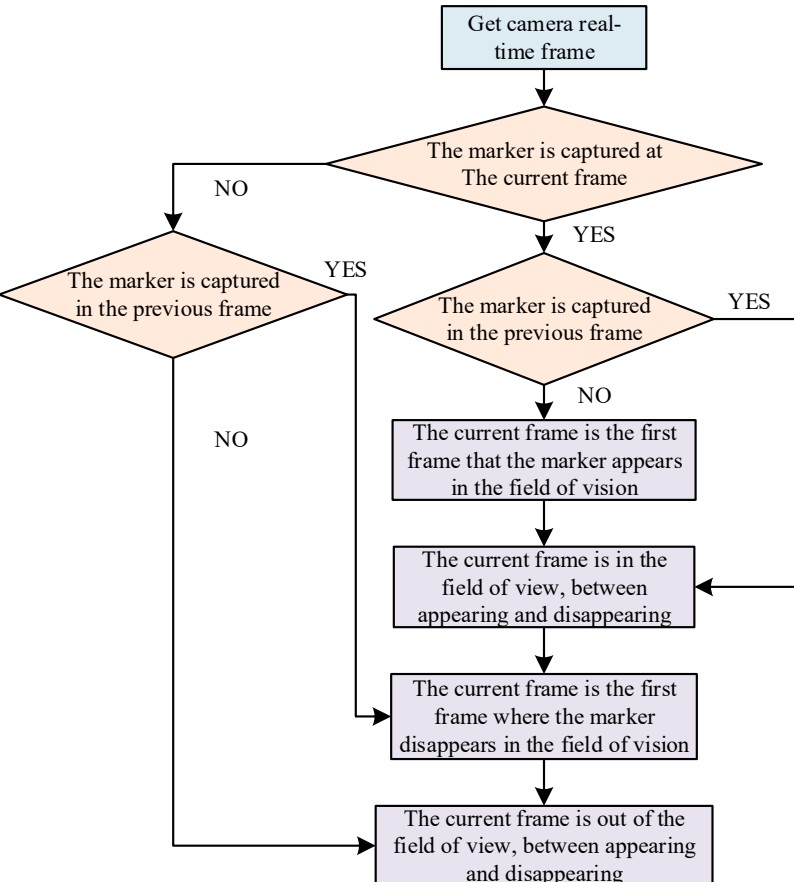

**Figure 3.** The judgment of the appearance and disappearance of markers.

In the formula, $n$ is the rotor speed, rpm. $f_0$ is the camera frame rate, fps. $M$ is the number of pictures collected by the camera in one rotation, frame. $f_p$ is the frequency of the

first frame of the marker in the field of vision, fps. $f_q$ is the frequency of the last frame of the marker in the field of vision, fps.

The judgment process for key frames is shown in Figure 3. Firstly, by determining whether the current frame has captured the marker, it is determined whether the current frame is within the visible range of the camera. Secondly, if the current frame captures the marker and the previous frame also captures the marker, then the current frame is a continuous frame within the field of view. If the marker was not captured in the previous frame, the current frame is the first frame where the marker was captured. Finally, if the marker is not captured in the current frame and was captured in the previous frame, then the current frame is the first frame where the marker disappears. If the marker is not captured in the previous frame, then the current frame is a continuous frame outside the field of view.

The period measurement method calculates the time of adjacent pulses, but this part of the time calculated by video speed measurement is easily affected by the frame rate. Only when the time of HGU rotor rotation is a multiple of the camera sampling period (the reciprocal of frame rate), can it be ensured that the time corresponding to the frame of adjacent markers appearing (or disappearing) is exactly the time required for rotor rotation. In order to reduce such errors, an improvement link is set up: when calculating the time corresponding to the adjacent frame of the marker, plus the time difference corresponding to the adjacent frame of the marker disappearing, the two calculation results are put into the speed list, which is helpful to reduce the calculation error caused by accidental factors.

In order to avoid the impact of random noise or camera frame leakage on the real-time rotor speed, the data will be further digitally filtered after the speed measurement by the period measurement method. In this paper, the median average filtering method is adopted, that is, the maximum and minimum values are removed from a set of numerical lists and the average value is taken, which is equivalent to "median filtering method" + "arithmetic average filtering method".

The speed measurement method based on image recognition technology is similar to the digital circuit speed measurement method, which is applicable to low speed measurement. Since the rotor speed for HGU under normal or abnormal states is not more than 150 r/min, it belongs to low-speed rotation, so the speed measurement method of the period measurement is more suitable for the RSM of HGU software.

### 3.2. Algorithm Steps and Processes

The process of rotor speed calculation in this paper can be divided into five parts: obtaining the HGU rotor video, dynamic capture of markers, rotor speed calculation, HGU operation state analysis, and returning the data to the background. The four parts are the main algorithms, and the principle is shown in Figure 4.

(1)  HGU rotor video: the real-time video of the HGU operation site is collected by the ip camera set around the water turbine. After the collection, the video is uploaded to the server for the next RSM preparation.

(2)  Dynamic capture of markers: the RSM of HGU depends on the setting of the markers. The real-time rotor speed of HGU can be calculated by dynamically capturing the markers in each frame of the video captured in step (1).

(3)  The rotor speed calculation of HGU: through the markers captured in step (2), find the key nodes, and the nodes where the markers appear and disappear, and measure the rotor speed by the period method.

(4)  Judge the operation state of the HGU: through step (3), the two final speed values of the two cameras are obtained, and the mean value of the two values is calculated, which is the real-time rotor speed of the HGU calculated by the program, so that the operation state of the HGU can be judged.

(5)  Return the data to the background: the server obtains the rotor speed of HGU, and finally sends the rotor speed information to the database according to the TCP com-

munication protocol and the video information according to the DUP communication protocol for storage.

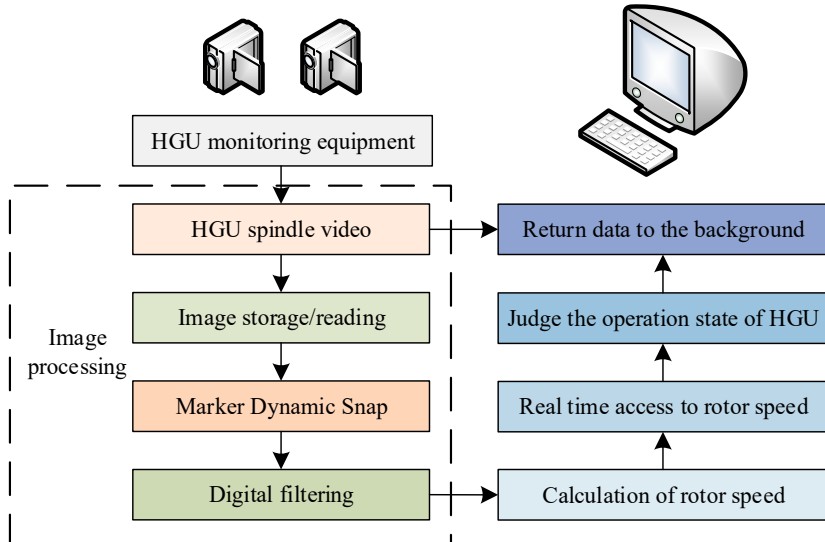

**Figure 4.** Speed measurement process.

## 4. YOLOv5 Test Result Analysis

### 4.1. Experimental Environment and Training Process

The parameters in the deep neural network mainly include the parameters that are automatically adjusted through learning and the hyperparameter that needs to be manually set. The adjustment of the hyperparameter is an important link between the theoretical knowledge of deep learning and the actual situation at present. The hyperparameter configuration of this training is as follows: the initial learning rate is $1 \times 10^{-4}$, the momentum is 0.0005, the batch size is 16, and the epoch is 200. The specific configuration of the computer is shown in Table 2.

**Table 2.** Computer configuration table.

| Device | GPU | NVIDIA GeForce GTX 2080 |
|---|---|---|
| Operating system | Operating system | Windows10 |
| | Computer language | Python3.6.12 |
| | Deep learning framework | Pytorch1.7.1 |

### 4.2. Data Set

YOLOv5, as a supervised learning algorithm, cannot be separated from the support of a large amount of data. The quality and distribution of data sets are important factors affecting the performance of the algorithm. The dataset used in this article originated from an actual hydropower plant and ultimately collected 2500 image data with a size of 640 × 480 in jpg format. The image data are divided into training set, validation set, and testing set in a ratio of 8:1:1.

In order to create a PASCAL VOC format dataset, LabelImg software was used to visually annotate each image. When annotating, the mouse was used to accurately and meticulously draw the border of the target as much as possible, which helps with the training and segmentation effect of the model. At the same time, the category name of the target is written on the border. As shown in Figure 5, it is the detection category "Mark" where the marker appears.

The result of image annotation is a Json format file, which is converted into a structured XML language based document, namely an xml format label file, which can better describe the target category, position, size and other attributes in the image.

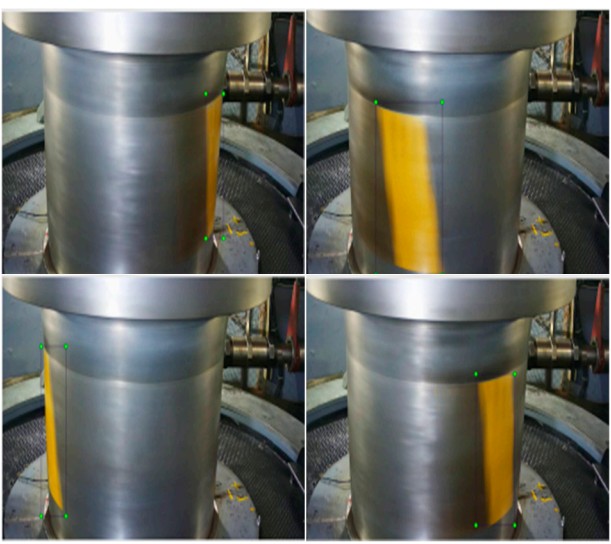

**Figure 5.** YOLOv5 picture annotation diagram.

### 4.3. Evaluating Indicator

This paper uses labeled turbine rotor images as training and validation sets and conducts 200 epochs of training in the YOLOv5s model to obtain the optimal model weight file. Figure 6 shows the training results of the turbine rotor markers in this model.

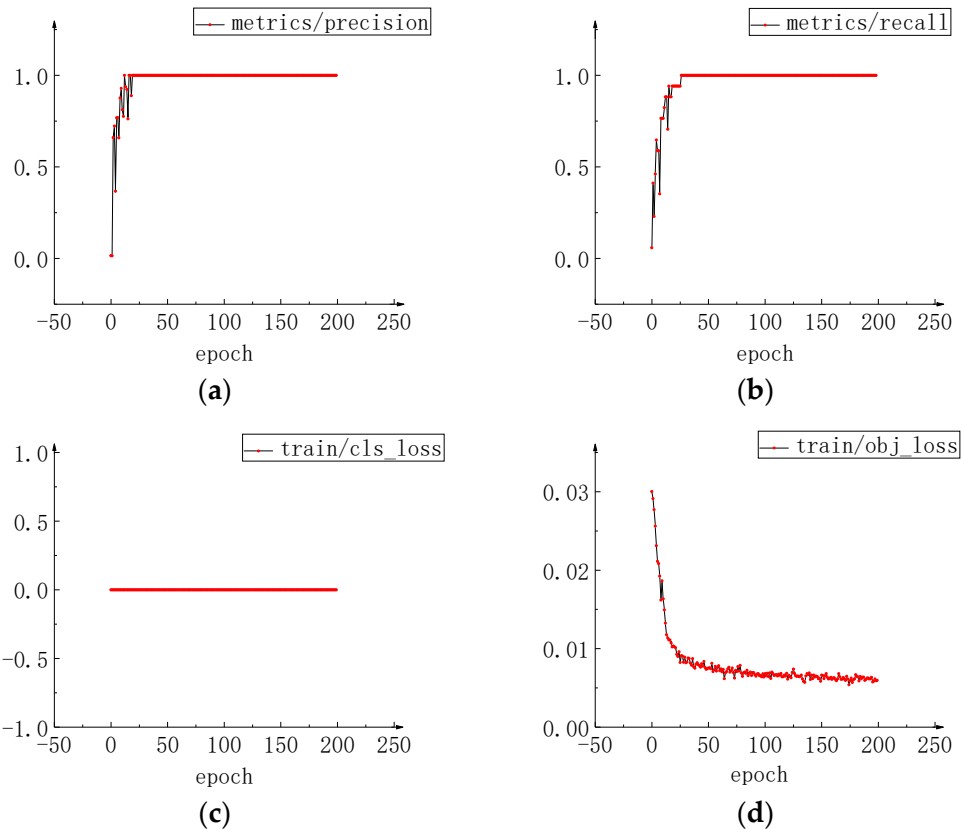

**Figure 6.** *Cont.*

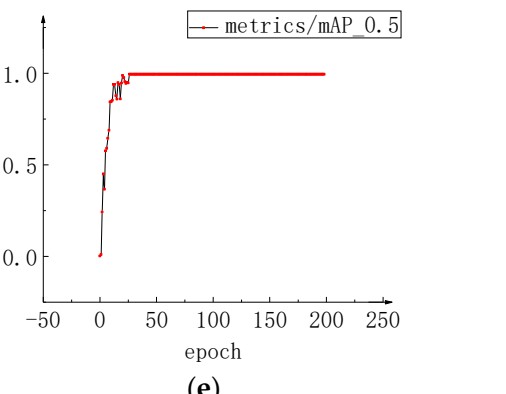 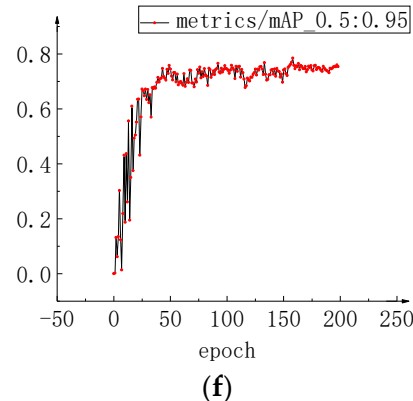

(**e**)　　　　　　　　　　　　　　　　　(**f**)

**Figure 6.** Analysis chart of training results. (**a**) Precision. (**b**) Recall. (**c**) Classified loss. (**d**) Confidence loss. (**e**) IOU = 0.5, mAP. (**f**) IOU ∈ [0.5:0.05:0.95], mAP.

From the figure, it can be seen that the detection Precision and Recall of YOLOv5 for rotor images have reached 99.5% and 100%, respectively, indicating that the model has strong generalization ability and robustness, can adapt to various complex scenes and environments, and can accurately locate and recognize markers for rotor images. At the same time, observing the changes in Classified loss and Confidence loss with the training process, it was found that after approximately 100 epochs, both tended to stabilize, indicating that the model had converged to a better state. From the fact that the Classified loss in the training set and the test set is zero, it can be concluded that the model can identify all target categories in the training set without classification errors, and the Classified loss of the model in the test set is also zero, indicating that the model has no overfitting or under fitting problems.

In addition, through the mAP indicators under different IOU thresholds, it was found that when IOU is 0.5, mAP approaches one, and when mAP@.5:.95, it is also close to 80%, indicating that the model maintains high detection performance even for smaller or more difficult to recognize targets. These indicators all demonstrate that the model has good detection performance for markers and meets the precision requirements for target recognition in RSM.

*4.4. Training Result Analysis*

In order to verify the effectiveness of the YOLOv5 algorithm for marker capture, YOLOv5 algorithm was compared with the traditional object capture algorithm based on histogram reverse projection. The detection results are shown in Figure 7. As can be seen from the left column of Figure 7, when using histogram reverse projection, due to the single projection sample that can be selected, it is difficult to regularly frame and select markers during the operation of the HGU. Moreover, when the color of the surrounding environment is similar to the color of the marker, noise is prone to occur, as shown in the red box selected area in the last group of comparison images, and cannot be used for subsequent RSM of HGU.

The comparison of object capture results based on YOLOv5 is shown in the right column of Figure 7. During the operation of the HGU, the rotor markers have undergone a certain degree of deformation, but since the markers with different degrees of deformation have been labeled during the labeling phase, the model can still accurately identify various forms of markers during the detection phase, that is, the Recall of the model for the markers is 100%. As the marker area decreases, the confidence level of target detection decreases slightly, laying a good foundation for subsequent velocity measurement.

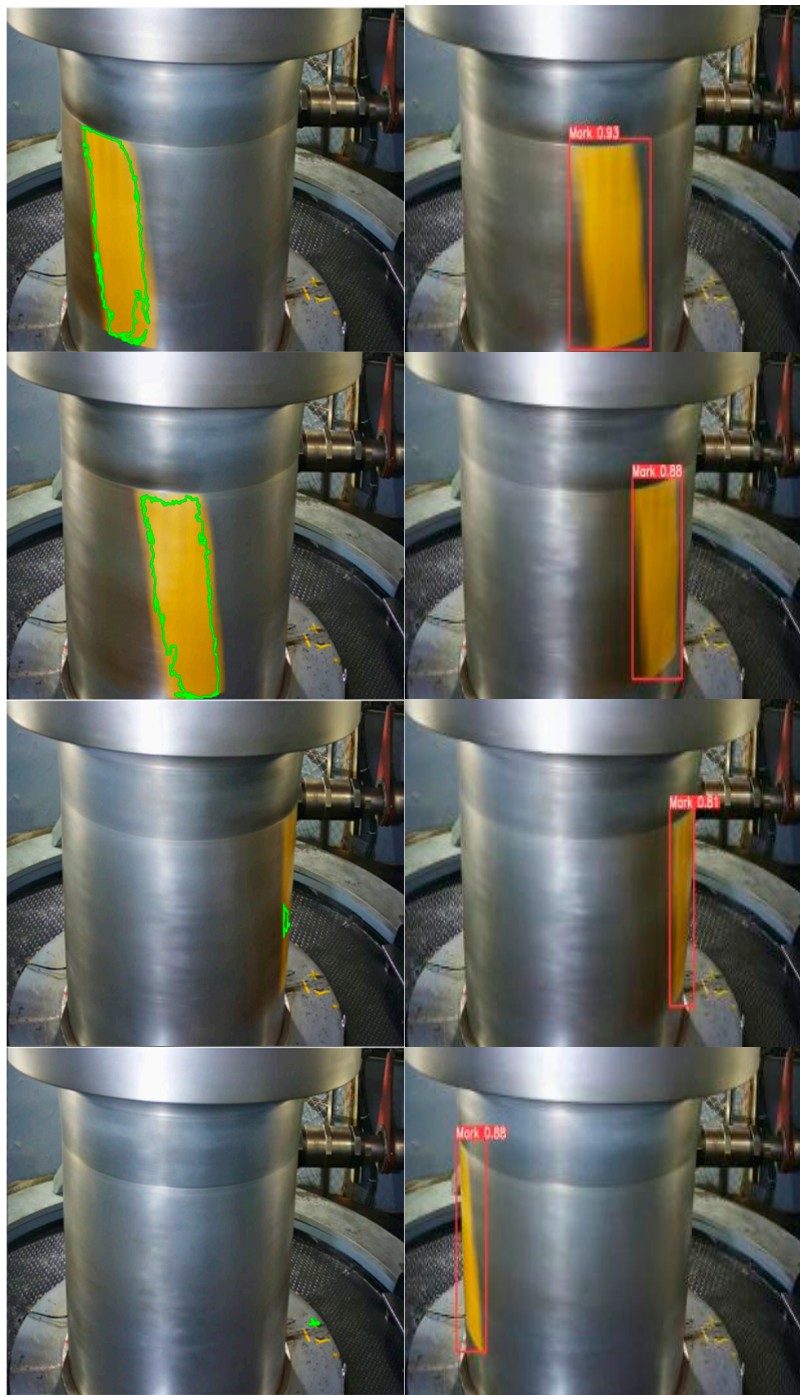

**Figure 7.** Comparison of YOLOv5 and histogram back projection detection results.

## 5. Field Measurement and Analysis

### 5.1. HGU State Definition

The test site is a hydropower plant in China, which is equipped with four units with a unit capacity of 200 MW and a total installed capacity of 850 MW. Based on the installed capacity and actual operation of the water turbine, different states of the HGU can be identified based on the RSM, as shown in Figure 8.

Firstly, the speed can be divided into a normal operating state and abnormal operating state. Secondly, during the startup state in normal operation, if the speed is greater than 95% of the rated speed, it is in the excitation state. During the shutdown state in normal operation, if the speed is less than 2% of the rated speed, it is considered a creeping state,

and if the speed is less than 25% of the rated speed, it is considered an air brake state. Finally, abnormal operating states include electrical overspeed when the speed exceeds 1.15% of the rated speed, and mechanical overspeed when the speed exceeds 1.35% of the rated speed.

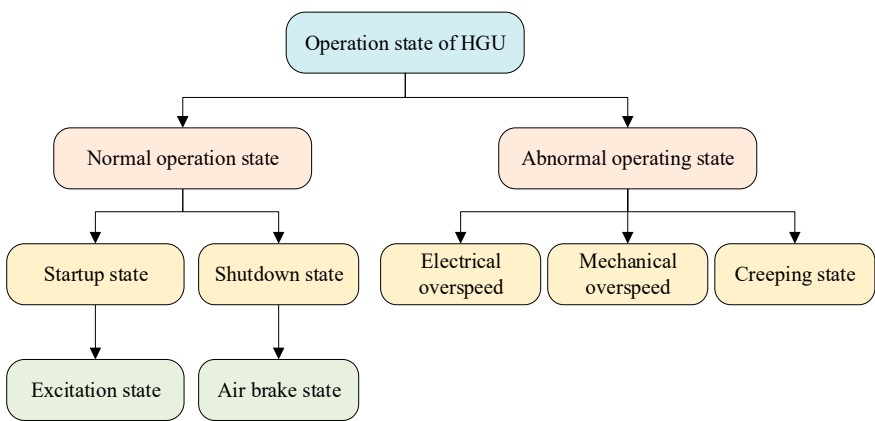

**Figure 8.** State of the HGU.

*5.2. HGU State Judgment*

The judgment of HGU state is an important content of HGU state monitoring. The HGU state not only depends on the numerical value obtained from the speed calculation, but also on the trend of the speed over time. For example, during the shutdown state, the unit may experience creep, and at this time, an air brake needs to be added to control the speed. During the startup state, the unit needs to input excitation to increase output power. In order to accurately determine the trend of rotational speed, a time window based method was adopted in the program. Specifically, a thread has been added to the program to record the speed $V_q$ one second ago. When calculating the real-time speed $V_p$ at the current moment, first compare it with $V_q$. If $V_p$ is smaller than $V_q$ and less than a preset fixed value (set to avoid data fluctuations causing misjudgment), then it can be considered that the unit is in a shutdown process, that is, the speed is gradually decreasing. On the contrary, if $V_p$ is greater or equal than $V_q$, then it can be considered that the unit is in the start-up process, that is, the speed is gradually increasing or maintaining stability.

The method proposed in this paper determines whether the HGU has stopped operation by measuring the area changes of the markers in consecutive frames of the video. As shown in Figure 9, firstly, a certain marker on the HGU is captured and tracked in real-time, and each video image is processed to extract the contour of the marker and calculate its pixel area S. Then, compare the area difference of the markers in two adjacent video images ($\Delta S = |S_1 - S_2|$). If $\Delta S$ is less than a given threshold, it is considered that the HGU is in a stationary state. Finally, when using binocular cameras, it is necessary to synchronize the video images collected by the two cameras and use an "OR" statement to perform logical operations on the static state collected by the two cameras. As long as one camera detects that the water turbine has stopped running, it is considered that the speed is zero.

The on-site test is carried out during the startup and shutdown test after the unit maintenance, which is helpful for the judgment of multiple states in a short time. Since there are no two states of overspeed in the field test, this test only records the following state judgments (both of them have been judged in the laboratory), as shown in Figure 10 below. It can be seen from the figure that when the HGU transits from the shutdown state to the speed-up phase, the rotor speed increases from 0 rpm to 104.778 rpm, and the unit is in the excitation stage. The rotor speed continues to rise until the rotor speed reaches the rated rotor speed of 107.067 rpm. After maintaining operation for a period of time, the HGU starts to decelerate, and the rotor speed drops to 68.873 rpm. The unit entered the stage of air brake to speed-up the shutdown process, and finally the rotor speed continued to drop to zero. However, due to the large number of guide vanes, it is impossible for the

guide vanes to be completely sealed under the closed state, and the water leakage of guide vanes is objective. When the water leakage increases to a certain extent, the water will impact the HGU runner, causing the rotating parts of the unit to produce slow rotating motion, and the rotor speed will reach 1.58 rpm from zero, which means the unit will appear in peristalsis.

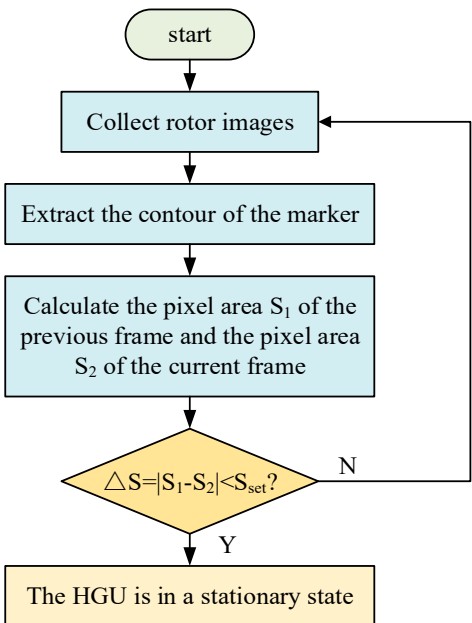

**Figure 9.** Flow chart of turbine shutdown state judgment.

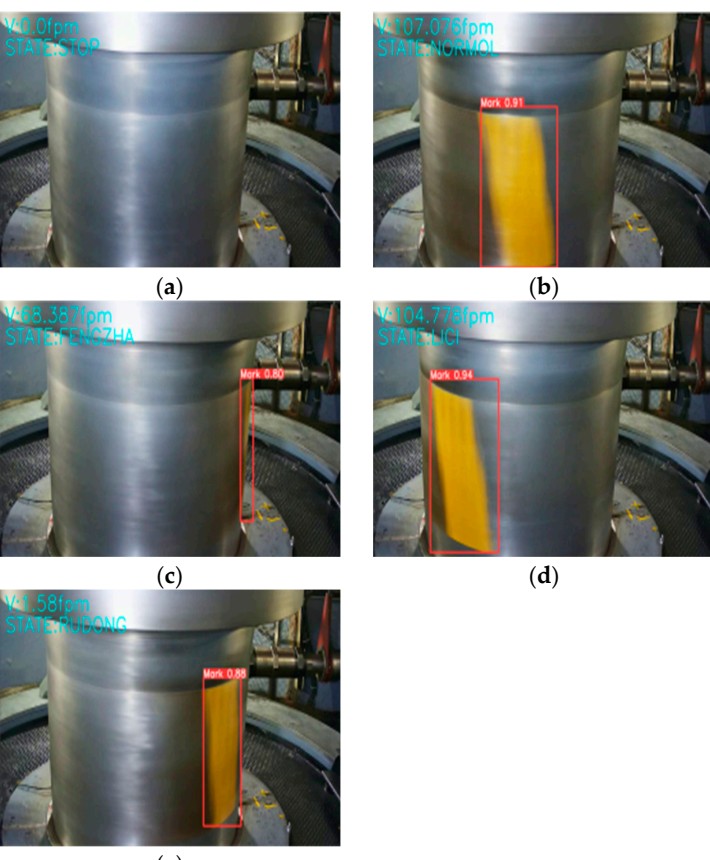

**Figure 10.** Unit state judgment diagram. (**a**) Shutdown state. (**b**) Excitation state. (**c**) Rated rotor speed state. (**d**) Air brake state. (**e**) Creeping state.

*5.3. Analysis of the Change of HGU Start and Stop Rotor Speed*

The rotor speed increased from 0 rpm to 107.1 rpm in the process of starting, and the starting time of this field test was about 180 s. In order to test the precision and tracking of the algorithm proposed in this paper, the real value of the rotor speed is recorded every five seconds and compared with the calculated rotor speed at the same time.

The overall change trend of rotor speed is shown in Figure 11a, with an average relative error of 2.45%. It can be seen from the figure that the overall following and precision of the calculated rotor speed are relatively high, but because the relative error is small, it is not convenient to analyze the error at different stages, so the low-speed stage, speed-up stage and stable stage of the rotor speed in Figure 11a are enlarged. It can be seen from the figure that, as shown in Figure 11c, due to the stable acceleration, the rotor speed in the speed-up stage is approximately linear, and the corresponding curve of the real value and the calculated value almost coincide, with high speed precision. In the low-speed stage (Figure 11b) and stable stage (Figure 11d), the calculated values are relatively small due to the slow speed change and the rotor speed is easily affected by historical data. It can be seen from the figure that the calculated value curve is below the true value curve. Finally, when the rotor speed is stabilized to the rated rotor speed, the calculated rotor speed will also tend to be stable.

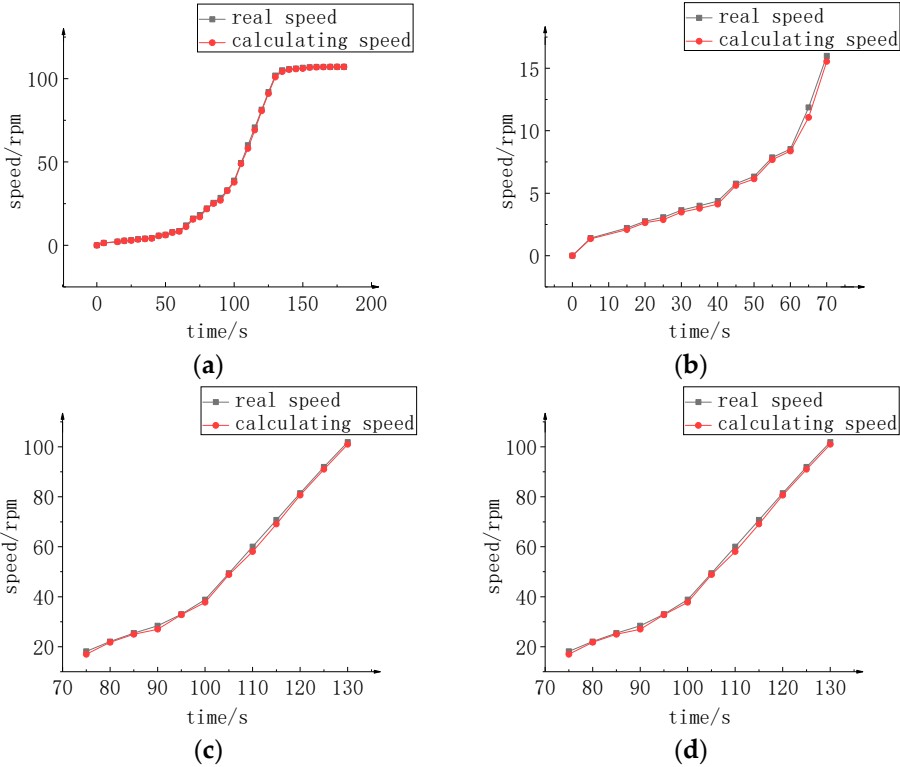

**Figure 11.** Rotor speed change diagram of HGU startup process. (**a**) Rotor speed change trend during startup process. (**b**) Rotor speed at low-speed stage. (**c**) Rotor speed at speed-up stage. (**d**) Rotor speed at stable stage.

During the shutdown process of the HGU, the rotor speed decreased from 107.1 rpm to 0 rpm. The shutdown test time was about 410 s, and the recording method was the same as that of the appeal startup process, which was recorded every 10 s. The overall trend diagram of the shutdown process is shown in Figure 12a. It can be seen from the figure that the precision of the calculated rotor speed is high and the tracking is good, with an average relative error of 2.44%. After the HGU is shut down, the guide vane is fully closed and the rotor speed decreases. However, because the HGU will damage the oil film of the bearing at low speed, resulting in excessive bearing friction, temperature rise and burning

loss, the air brake must be put into operation to shorten the time of the unit at low speed, quickly stop rotating, and protect the HGU. From the Figure 12b–d, it can be seen that the calculated rotor speed are mostly above the true value, that is, slightly higher than the true value. This is mainly because the historical rotor speed during the shutdown process is relatively large, which affects the true value in the digital filtering stage. This law is just opposite to the startup process.

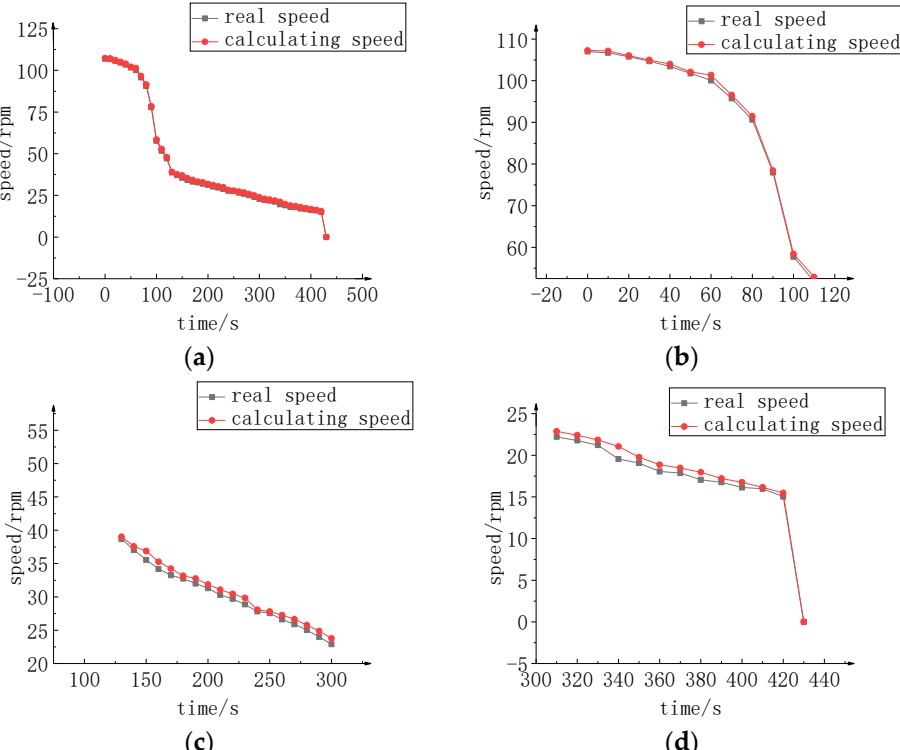

**Figure 12.** Rotor speed change diagram of HGU shutdown process. (**a**) Rotor speed change trend during shutdown process. (**b**) Rotor speed at low-speed stage. (**c**) Rotor speed at speed-up stage. (**d**) Rotor speed at stable stage.

### 5.4. Display of Monitoring System

This paper unifies the deployment of hardware devices and detection algorithms, and develops a monitoring system for the rotor speed of HGU based on mobile applications. The application of the HGU rotor speed monitoring system for continuous operation monitoring of a HGU rotor in operation, and the experimental results are shown in Figure 13.

The main functions of the HGU rotor speed monitoring system include calling a camera for real-time state monitoring, outputting algorithm recognition results for each stage of the rotor, judging the current state of the rotor, and summarizing and recording dimensional information such as time, state, and unit number, for users to view the historical state of the unit at any time. Based on the algorithm proposed in this paper, the system can effectively determine the rotor speed and current state of HGU, and the specific detection results are shown in Figure 14.

Figure 14 shows the judgment results of the system, where Figure 14a–f show the six operating states of the HGU, namely normal state, excitation state, acceleration state, air brake state, deceleration state, and shutdown state. After testing, the monitoring system can accurately identify the different operating states of the HGU.

The judgment results of the HGU rotor speed monitoring system are shown in Table 3. From the table, it can be seen that the speed accuracy measured by the system is relatively high, and the judgment of the unit's operating status is accurate.

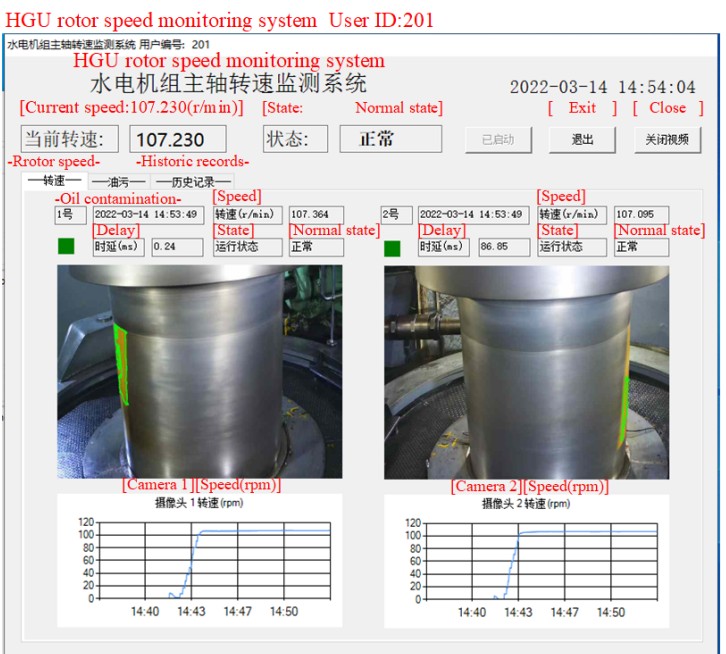

**Figure 13.** The experimental results of HGU rotor speed monitoring system.

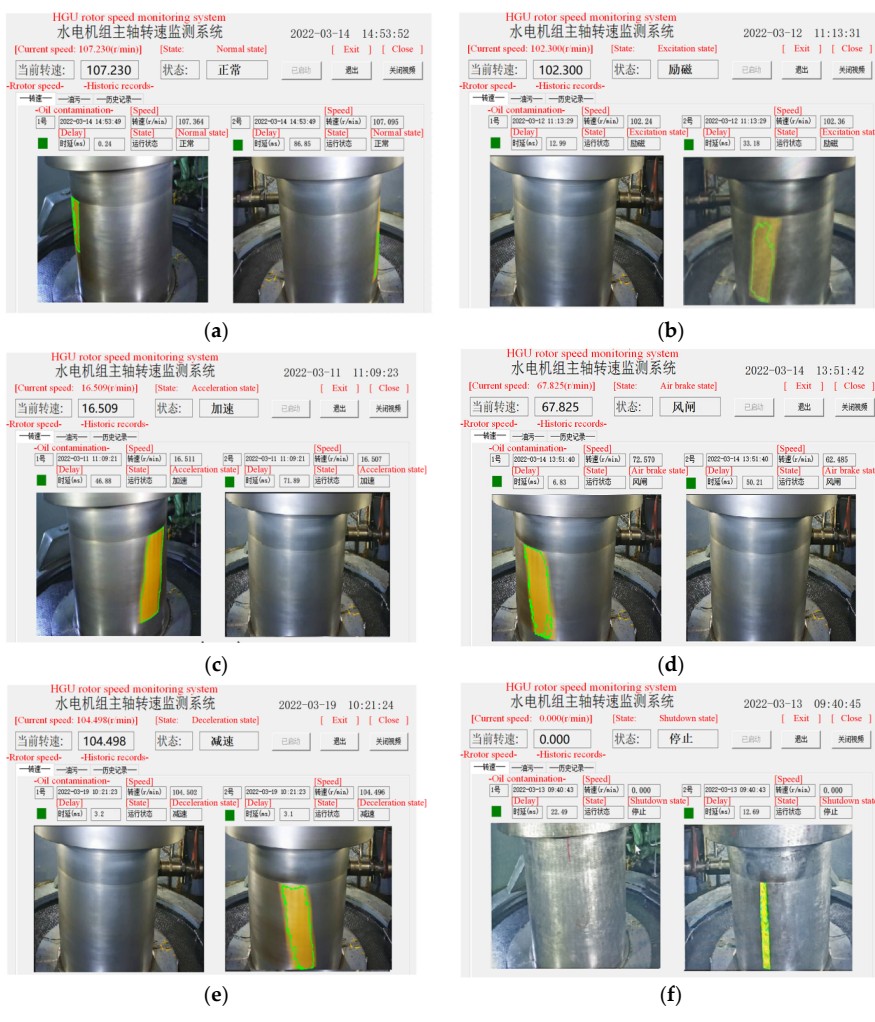

**Figure 14.** State determination results. (**a**) Normal state. (**b**) Excitation state. (**c**) Acceleration state. (**d**) Air brake state. (**e**) Deceleration state. (**f**) Shutdown state.

**Table 3.** Judgment results of the rotor speed monitoring system for HGU.

| Rotor Speed | Judged State | Actual State | Is It Accurate? |
| --- | --- | --- | --- |
| 107.230 | Normal state | Normal state | Yes |
| 102.300 | Excitation state | Excitation state | Yes |
| 16.509 | Acceleration state | Acceleration state | Yes |
| 67.825 | Air brake state | Air brake state | Yes |
| 104.498 | Deceleration state | Deceleration state | Yes |
| 0.000 | Shutdown state | Shutdown state | Yes |

The monitoring system can export the rotor speed change curve of the HGU starting and stopping, as shown in Figure 15. The horizontal axis represents time, and the vertical axis represents the rotor speed measured by the system.

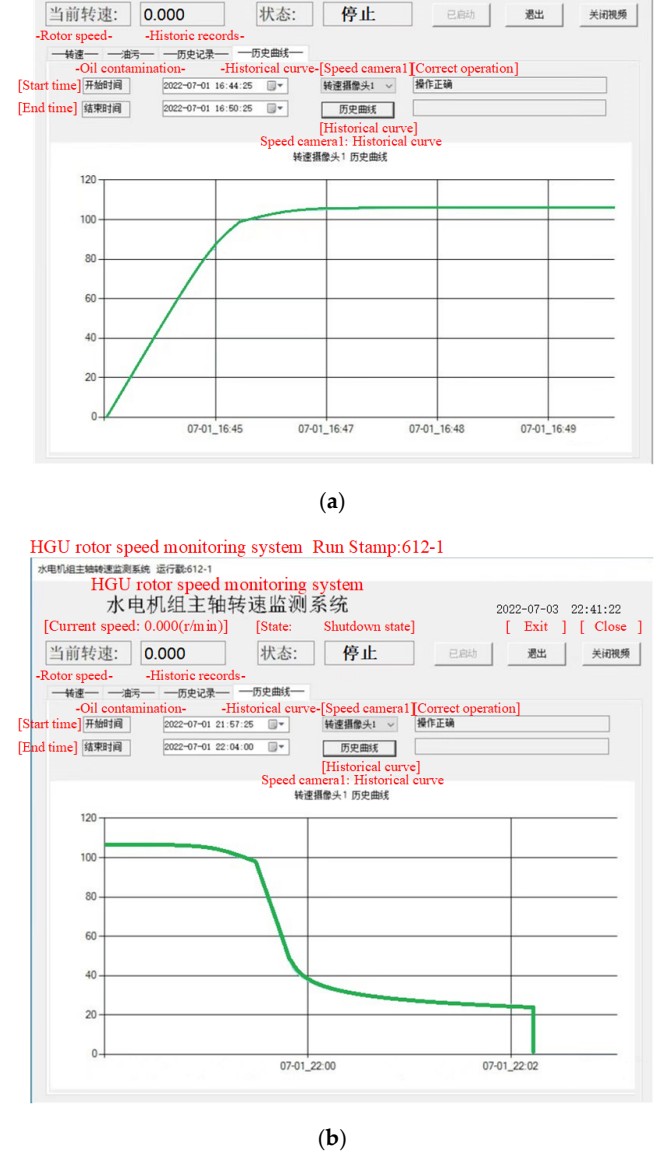

(**a**)

(**b**)

**Figure 15.** Startup and shutdown process curve of HGU. (**a**) startup process curve of HGU. (**b**) shutdown process curve of HGU.

The monitoring system can store daily data in detail and accurately and display the weekly or monthly data change trend in the data analysis module through the line chart.

The early warning module in the monitoring system can obtain abnormal data and establish a dataset to be transmitted to the database. The fault diagnosis module can identify abnormal data and diagnose faults with an accuracy rate of over 98%, and can provide accurate diagnostic reports. The effectiveness of the system has been verified through the application of the HGU rotor speed monitoring system for hydropower plants.

The performance comparison between the proposed method and the classical method in this article is shown in Table 4. From the table, it can be seen that the method proposed in this paper can achieve low-cost, noncontact, and high anti-interference RSM while ensuring accuracy, which has greatly improved performance compared to classical RSM methods.

**Table 4.** Performance comparison of different rotor speed measurement methods.

| RSM Method | Precision | Additional Equipment Configuration | Contact Measurement | Anti-Interference |
|---|---|---|---|---|
| Toothed disc | 0.001 | Yes | Yes | High |
| PT residual pressure | 0.01 | Yes | Yes | Low |
| Photoelectric encoder | 0.001 | Yes | Yes | Low |
| Laser Doppler | 0.01 | Yes | No | Low |
| The method proposed in this paper | 0.001 | No | No | High |

## 6. Conclusions

This paper proposes a method for RSM and OSI of HGU based on the YOLOv5. First, the YOLOv5 model is used to accurately capture the HGU rotor, then the period method is used to calculate the speed, and finally the operation state of the HGU is judged by the calculated speed. The method in this paper used to measure the speed of the HGU, which realizes the long-distance, noncontact and high-precision RSM of HGU, helps the staff better grasp the real-time operation of the HGU and makes the hydropower plant more intelligent and efficient. The specific performance is as follows:

(1) Using binocular camera to photograph the rotor can ensure real-time monitoring of the rotor.
(2) Accurately judging several different states of the HGU can help the staff find potential safety hazards quickly and on time according to the alarm prompt information, improve work efficiency and reduce labor costs.
(3) When the unit speed changes, the software can quickly follow its changing trend while maintaining the precision.

**Author Contributions:** Conceptualization, J.S. and J.L.; methodology, L.X., H.L. and J.L.; software, Python; validation, J.S., Y.L., R.Z. and J.L.; formal analysis, J.S. and L.X.; data curation, J.S., Y.L., R.Z. and L.X.; writing—original draft preparation, J.S., L.X., R.Z. and J.L.; writing—review and editing, J.S. and J.L.; project administration, J.L.; funding acquisition, J.L. All authors have read and agreed to the published version of the manuscript.

**Funding:** This research was funded by the Key R & D Program of State Grid Shaanxi Electric Power Company (SGTYHT/21-JS-223) and National Natural Science Foundation of China (52077176).

**Data Availability Statement:** Not applicable.

**Acknowledgments:** At the point of finishing this paper, I'd like to express my sincere thanks to all those who have lent me hands in the course of my writing this paper. First and foremost, I would like to thank my mentor Liu for his careful guidance at every stage of thesis writing. Secondly, I would like to thank my research group, which not only gives me economic support, but also gives me a perfect simulation platform. Last but not the least, I'd like to thank those leaders, teachers and working staff especially those in the Xi'an University of Technology. Without their help, it would be much harder for me to finish my study and this paper.

**Conflicts of Interest:** We would like to submit the enclosed manuscript titled "A Method for Rotor Speed Measurement and Operating State Identification of Hydro-generator Unit Based on YOLOv5", which we wish to be considered for publication in Machines. No conflicts of interest exit in the submission of this manuscript, and manuscript is approved by all authors for publication. I would like to declare on behalf of my co-authors that the work described was original research that has not been published previously, and not under consideration for publication elsewhere, in whole or in part.

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
