# Peer review of "A Method for Rotor Speed Measurement and Operating State Identification of Hydro-Generator Units Based on YOLOv5"

_machines, doi:10.3390/machines11070758_

Round 1

Reviewer 1 Report

This paper proposes a method for RSM and OSI of HGU based on the YOLOv5. The idea of using video image to realize the estimate of rotor speed is interesting. Some concerns are listed as follows.

1. Rotary speed measurement is not a new thing. It is necessary to provide more references of the existing works. Then the main deficiencies of them can be summarized.

2. Since the speed is determined by the marks captured by the current frame or the last frame. How to deal with such issues if the YOLOv5 fails to detect the marks of the current frame or the last when the model is not well trained.

3. To train the YOLOv5, a large amount of labeled data is necessary. Can the proposed method be directly used for speed measurement of a new equipment?

4. Please add the descriptions related to the main advantages and disadvantages of the proposed method in comparison with existing techniques.

Minor editing of English language is required to improve the quality of the manuscript.

Author Response

Dear Reviewer #1:

We are very grateful to the reviewers for their careful review of the paper and for their suggestions, which are very professional and valuable. We have revised your questions one by one. The detailed modification results and answers to the questions are attached for your reference. Thank you again for your recognition of our work and suggestions, which greatly improved our paper. It is very enjoyable to discuss the issues in the paper or future research directions with the reviewer. If this paper can meet your requirements, it will be the best. If there are still any questions in the future, we would be honored if you could also raise them.

Reviewer 2 Report

Dear Authors,

in my opinion, the submitted paper is very good. I don’t have remarks on this paper.

The paper presents the application of YOLOv5 for the rotor speed measurement and operating state identification of the hydro-generator unit.

I consider the topic relevant in the field because it presents application of the vision artificial intelligence for the rotor speed measurement.

The paper addresses the usage of vision artificial intelligence for the rotor speed measurement 

The conclusions are consistent with the rest of the paper.

The references are appropriate.

Author Response

Dear Reviewer #2:

Thank you for your affirmation of our paper. We will continue to work hard.

Reviewer 3 Report

It should be great if the authors give comparison to other realizations presented in the literature.  

Author Response

Dear Reviewer #3:

We are very grateful to the reviewers for their careful review of the paper and for their suggestions, which are very professional and valuable. We have revised your questions one by one. The detailed modification results and answers to the questions are attached for your reference. Thank you again for your recognition of our work and suggestions, which greatly improved our paper. It is very enjoyable to discuss the issues in the paper or future research directions with the reviewer. If this paper can meet your requirements, it will be the best. If there are still any questions in the future, we would be honored if you could also raise them.

Reviewer 4 Report

The introduction contains a sufficient number of references but all the references are 2022-2023, only 3 before, which shows the research is not compared with classical methods. 

The introduction is very technical and you don't manage to get an idea of what is being pursued with the application of artificial intelligence. Artificial intelligence is not presented except through the YOLO software.

Can't we follow a thread of comparative research with classical measurements and what are these that have been used so far?

The algorithm that makes the innovation of the proposed research is not highlighted.

Moreover, it's just a determination of the images from a film and we don't know if the result is the one taken or if it's wrong compared to the standard.

Certain terms are defined (Precision, Recall, IOU, etc.) which have certain values but are not defined as limited between which they must be situated in order to know the result of this good.

3.2. Algorithm steps and processes is too superficially presented.

For Graphs present it will be good to create a table and make parallels between values and come up with a conclusion reflect by graphs.

From figure 13 - 15, it will be good to present the translation of language because only some readers will understand for what are the values.

The conclusion it will be good to present some percentage of what the algorithm realizes in comparison with classical methods

The research paper is much more dedicated to specialists in the domain of software then the general public of machines journal in my opinion.

Author Response

Dear Reviewer #4:

We are very grateful to the reviewers for their careful review of the paper and for their suggestions, which are very professional and valuable. We have revised your questions one by one. The detailed modification results and answers to the questions are attached for your reference. Thank you again for your recognition of our work and suggestions, which greatly improved our paper. It is very enjoyable to discuss the issues in the paper or future research directions with the reviewer. If this paper can meet your requirements, it will be the best. If there are still any questions in the future, we would be honored if you could also raise them.

Round 2

Reviewer 1 Report

The authors have addressed all the issues.

None